# Research on the Damage Law and Prevention Measures of the Substrate under the Action of Water and Rock

**Qiushuang Zheng [1] and Lifu Pang [2,3,*]**

1   School of Economics and Management, Beijing University of Posts and Telecommunications, Beijing 100876, China
2   College of Energy and Mining Engineering, Shandong University of Science and Technology, Qingdao 266590, China
3   State Key Laboratory of Mine Disaster Prevention and Control, Shandong University of Science and Technology, Qingdao 266590, China
*   Correspondence: plf1715@163.com

**Abstract:** The potential of sudden water accidents induced by damage to the bottom slab that forms a water channel due to the action of karst water calls for research on the damage law of coal seam floors under the coupling effect of mining and karst water. In this study, the coupling situation of karst water and floor rock was analyzed based on the theory of rock mechanics and fracture mechanics, combined with the actual geological conditions of a coal seam floor. The law of water-rock coupling was investigated using theoretical analysis and mechanical tests. Results indicate that the strength of the bottom slab rock decreased significantly when the karst pore water was coupled with the bottom slab rock. A three-dimensional numerical model was established using FLAC3D software to simulate the mining situation of the working face under different water pressures. By analyzing the stress change, water pressure change, and plastic zone damage of the bottom slab, it was discovered that the damage effect of karst water pressure on the bottom slab continues to increase when the water pressure is greater than 3 MPa. The impact caused by water pressure greater than 6 MPa occurs gradually. Analysis of measures to prevent and control sudden water on the bottom slab indicates that grouting reinforcement can effectively prevent and control the bottom slab karst sudden water problem.

**Keywords:** water-rock coupling; subgrade damage; hydraulic pressure; numerical simulation; prevention and control measures

## 1. Introduction

With the development of coal mining technology and detection technology and the gradual concentration of coal seams in deep areas, the threat of shallow surface water hazards and traditional mining water hazards has gradually decreased [1–3]. The main threat in deep mining processes comes from floor karst water. Floor karst water is in the original equilibrium state together with the floor strata under hydrostatic pressure. In mining when the original equilibrium state is broken, hydrostatic pressure becomes the dynamic water pressure, and the water pressure is in a state of change, the bottom aquifer will be damaged with the possibility of flooding [4–6]. At present, domestic and foreign experts and scholars have carried out some research on the mechanism and law of floor water inrush caused by karst water. Based on numerical simulation software, some experts and scholars have carried out research on floor failure and fluid-solid coupling [7–10].

Zhang Peisen et al. adopted the fluid-solid coupling model FLAC3D to simulate the entire process of coal mining and proposed that the coupling effect of mining stress and water pressure causes the initiation, expansion, and penetration of cracks in floor rock strata as well as water inrush [11]. Sun Jian et al. established a three-dimensional fluid solid coupling model through secondary development of FLAC3D software. From the

perspective of numerical simulation, the relationship between the change of confined water pressure in the floor of inclined coal seams and the threat of water inrush is explained as mining proceeds [12]. Ma Changqing and others used numerical simulation software in combination with on-site measurement to conduct a study on the failure and deformation of surrounding rock and elaborated on the deformation of surrounding rock caused by mining [13]. Based on crack propagation and combined with rock damage theory, some experts and scholars have conducted research on rock failure laws under water rock action. Gao Saihong et al. considered the damage and fracture mechanism of fractured rock mass under water action and deduced the stress intensity factor of cracks under water action [14]. Odintsev and Miletenko established a confined water pressure fracturing model and found that natural water pressure fracturing is constrained by natural and induced stresses, groundwater hydrostatic pressure, and mining sequence [15].

Based on on-site measurement and monitoring through technical means, some experts and scholars have conducted research on the monitoring of karst water hazards and the formation of water-conducting fracture zones. Shi Longqing et al. used a nonlinear risk assessment method to evaluate and analyze water inrush from the coal seam floor, optimizing the inaccuracy of the water inrush coefficient method [16]. Liu Weitao and others used on-site monitoring equipment and theory to calculate the development height of floor cracks, revealing the development law of floor cracks [17]. Sun Yunjiang et al. used an ESG microseismic monitoring system to conduct real-time monitoring of the formation process of water-conducting fracture zones on the floor, revealing the relationship between fault fracture expansion and confined water pressure [18].

In the above research, numerical simulation, field measurement, and experimental methods were involved for the floor water inrush disaster. The above studies lack the characteristics of continuous floor failure, water pressure changes, in situ stress changes, and stress failure caused by rock water pressure. Therefore, this article comprehensively combines theoretical analysis, mechanical tests, and numerical simulation to systematically carry out research on the prevention and control of floor water hazards under the action of water and rock.

## 2. Analysis of Water-Rock Coupling Law in Deep Substrate

### 2.1. Analysis of the Mechanism of Water-Rock Action

According to the actual geological conditions of coal mining combined with "Four-belt" mining theory, the mining process of confined water in the bottom strata is divided into the upper failure zone, the middle intact zone, the middle and lower semihumid zone, and the lower complete water-rock coupling zone (see Figure 1).

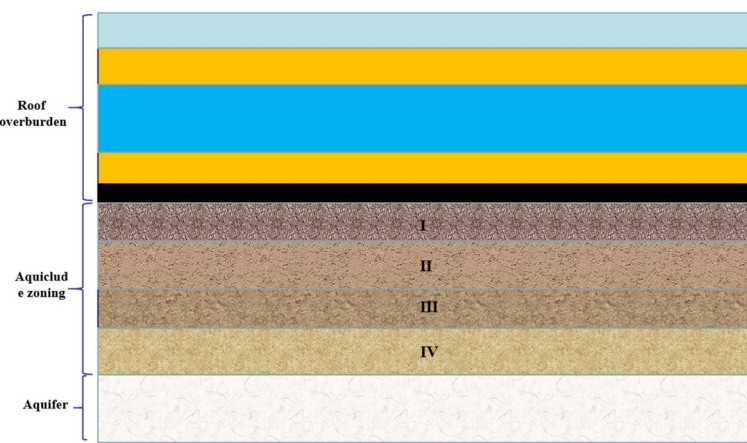

**Figure 1.** Different partitions of the water barrier.

The upper damage zone is in direct contact with the coal seam and is directly damaged by mining, and the upper damage zone appears to be a broken, fractured, plastic damage

area and is subject to stress concentration. The middle intact area is mainly subjected to the bottom plate stress transfer effect and to the mining disturbance effect; this area is generally subject to the limited disturbance effect, and the corresponding damage will occur under the effect of continuous strong mining disturbance [19–21]. The middle and lower areas are generally close to the location of the pressurized water, as the rock contains pores with permeability; pressurized water under the action of mining disturbance water pressure increases, as does permeability. The lower area of the water barrier is in contact with the aquifer, is subject to the weakening effect of water and water pressure, and is in a water-rock coupling state, and this part of the rock has saturated water rock and half-saturated water rock [22–25].

The above division is directly damaged in the upper area by mining activities, mainly by compression shear damage as shown in Figure 2, which is simplified to a uniform load q. Vertical cracks, inclined cracks, and herringbone cracks appear under pressure as shown in the Figure 2a–c. Starting from the middle region of the bottom rock layer, the karst water pressure exerts a pressure effect on this part. This part affected by mining disturbance also has the partial action of pressure-bearing water pressure, and its force is shown in Figure 3. The region beneath the lower portion of the slab in direct contact with karst water experiences slow seepage due to hydrostatic pressure. However, when mining activities occur, the water pressure changes, and this region is further disturbed, leading to the generation of numerous secondary fractures from the primary fractures. The karst water penetrates through the lower area of the substrate under water pressure, leading to complete water-rock coupling.

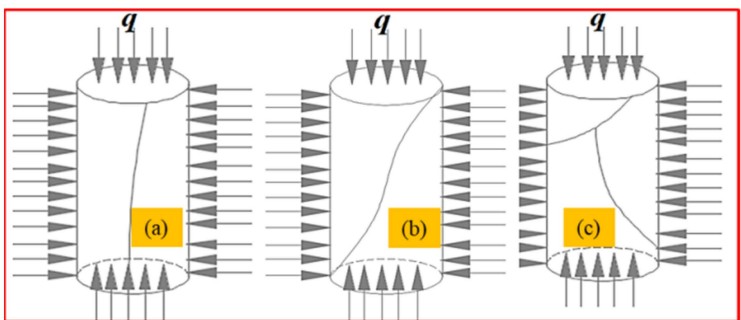

**Figure 2.** Different failure modes.

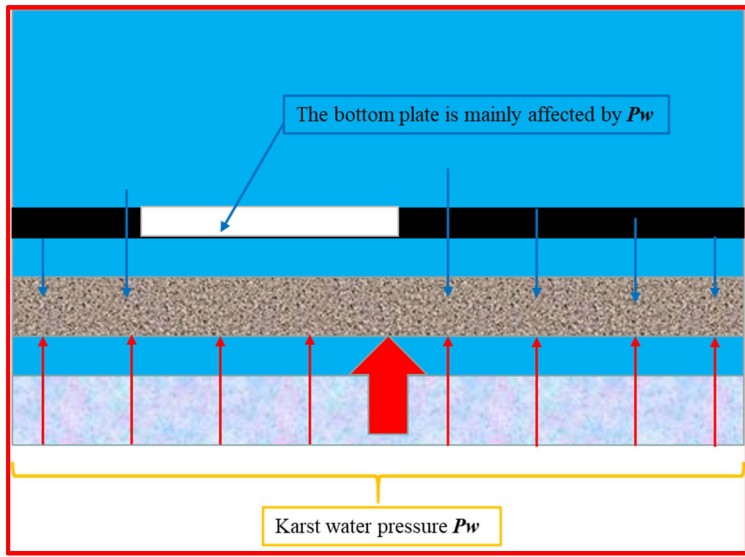

**Figure 3.** Mechanical model of the bottom slab subjected to karst water pressure.

According to the research of many rock mechanics experts, as long as there are connected fractures or pore systems in the rock, the law of effective stress in Taishaji can be applied. Therefore, the effective stress calculation Formula (1) can be obtained in the presence of pore water pressure,

$$\sigma^1 = \sigma - p \tag{1}$$

where, $\sigma$ is the total stress (MPa), $p$ is the pore water pressure (MPa), and $\sigma^1$ is the effective stress (MPa).

According to the Mohr-Coulomb strength theory, the shear strength of the rock formation in the lower part of the bottom slab in the area of complete water-rock action coupling under the action of pore water pressure is as follows:

$$\begin{cases} \tau_f = c + \sigma^1 tg\varphi \\ \tau_f = c + (\sigma - p)tg\varphi \end{cases} \tag{2}$$

According to Formulas (1) and (2), the pore water pressure in the floor rock will cause a decrease in rock stress strength. The greater the water pressure is, the more the rock strength will decrease. When the water pressure is close to 0, the rock strength will not be affected.

### 2.2. Conventional Triaxial Compression Test of Dry and Saturated Rock

According to the previous theoretical analysis, it is assumed that the strength and damage degree of the rock mass will be weakened under the coupling of water and rock. In order to accurately grasp the strength failure characteristics of rock mass under water pressure coupling, pseudo-triaxial compression tests under saturated and dry conditions were carried out using triaxial compression equipment. RLJW-2000 rock rheometer from the Mine Disaster Prevention and Control Laboratory of Shandong University of Science and Technology was selected as the test equipment, as shown in Figure 4. The conventional pseudo-triaxial compression test was realized by using the strain gauge, thermoplastic pipe, and sealed chamber shown in Figure 4. The rock specimen is sandstone and was processed into a standard cylindrical specimen with a size of $50 \times 100$ mm (diameter and height). The test was divided into two groups: the dry rock sample group and the saturated rock sample group. The former is the control group, and the latter is the test group. Before the test, the rock specimen was dried in an oven at 30 °C for 48 h, then the saturated rock specimen was soaked in distilled water and weighed every 12 h until the mass did not change for three consecutive times, reaching the saturated state.

According to the test results, the stress curves of rock samples under dry and saturated conditions were drawn as shown in Figure 5. According to Figure 5, the peak strength under the dry condition is 68 MPa, and the peak strength of rock under the saturated water condition is 43 MPa. Compared with the dry rock specimen, the peak strength decreases by 36% on average. This result strongly verifies the previous theoretical analysis. From Figure 5, it can be seen that the strength-change laws of dry rock specimens and saturated rock specimens are similar at the initial stage of loading. With the loading process, the stress curve of dry rock shows a linear elastic growth law, while the growth curve of the saturated rock specimen is slow. After reaching peak strength, the stress strength of both dry and saturated rock specimens decreased rapidly, and eventually there was definite residual strength [25,26]. The above research can well explain the failure mechanism of the coal seam floor under the interaction of water and rock. The rock at the junction between the lower part of the coal seam floor aquiclude and the aquifer can be considered to be in a saturated state. The increase in pressure of the confined water on the aquiclude decreases the strength of the aquiclude, and at the same time the water seeps into the floor aquiclude fractures, causing expansion of the fractures. On the other hand, during the seepage process, water has a weakening effect on the floor strata, resulting in a decrease in the strength of the impermeable layer.

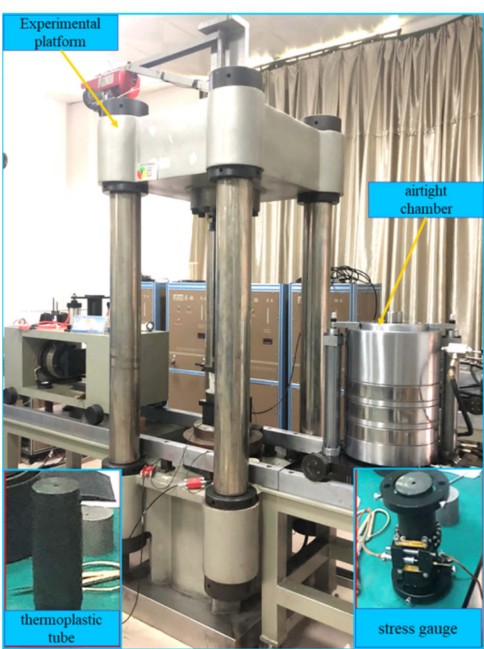

**Figure 4.** Experimental instrument.

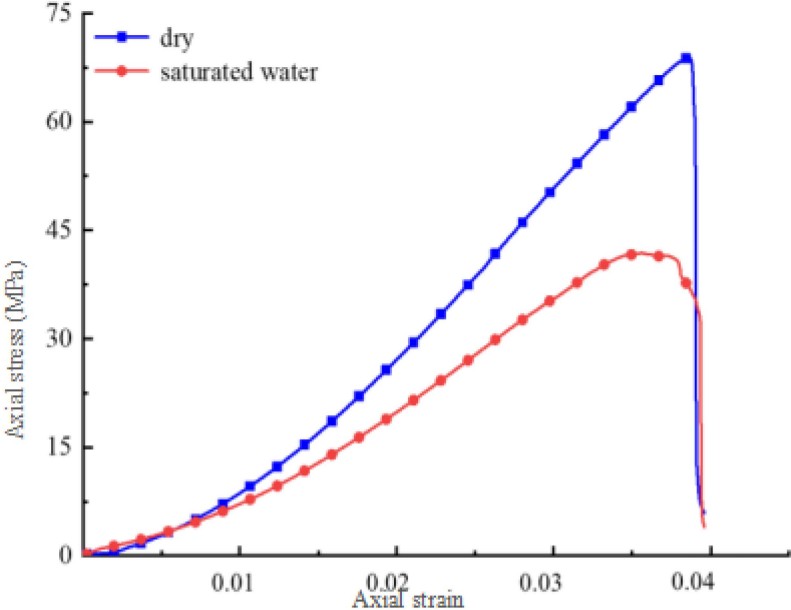

**Figure 5.** Stress-strain curves of dry and saturated rocks.

The failure forms of the dry and saturated rock test pieces are shown in Figure 6. Figure 6a shows the dry rock specimen after failure, and Figure 6b shows the saturated rock specimen after failure. The failure forms of the two groups of figures are compression-type failure cracks, and the crack angle and vertical and cross states can be seen.

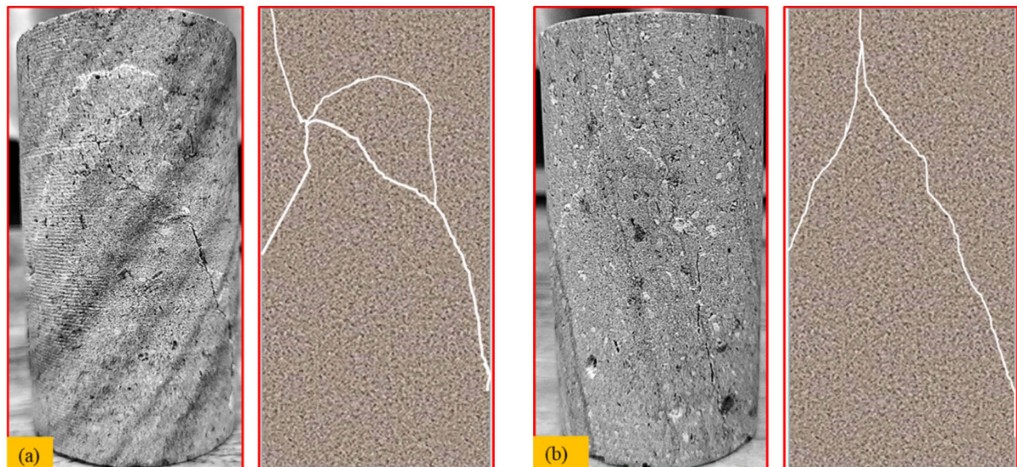

**Figure 6.** Specimen of failure characteristics. (**a**) Dry rock specimen after failure, (**b**) saturated rock specimen after failure.

To sum up, it can be analyzed that the rock strength decreases significantly under the coupling effect of water and rock, and the failure characteristics also conform to the theoretical analysis.

## 3. Numerical Simulation of Floor Failure under Confined Water

### 3.1. Model Establishment

Using the 1311 working face of Yangcheng coal mine as the geological background, a 3D geological model was established (Figure 7). Based on actual geological data, the length of the model is 200 m, the width is 200 m, the height is 110 m, and the thickness of the coal seam is 5 m. The rock mechanical parameters were obtained from mechanical tests (see Table 1). The model is divided into 36,800 units and 40,344 nodes. The front and back left and right boundaries of the model are set as horizontal direction constraints, the upper boundary is free boundary, and the bottom boundary is fully constrained; each excavation step is 10 m along the strike length, and there are 10 excavation steps in total. Considering the boundary effect, the distance of the coal pillars left and right from the working face opening and stopping line is 50 m, and the distance of coal pillars protected by the roadway is 50 m from the front and back boundaries of the model. In the simulation process, the Mohr-Coulomb plasticity principal model and Mohr-Coulomb damage criterion were used to calculate the mining damage characteristics of the coal seam floor. According to the experimental principle of the controlled variable method, the water pressure was set as a single variable, the water pressure range was 0–10 MPa, and other conditions were kept constant.

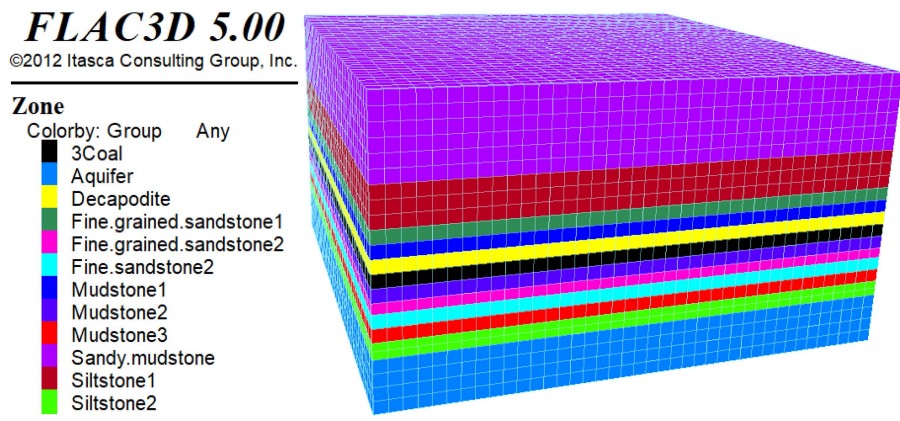

**Figure 7.** Three-dimensional geological model.

**Table 1.** Physics mechanics parameter of model terrane.

| Layer Group | Thickness/m | Density/(g·m⁻³) | Bulk Modulus/GPa | Shear Modulus/GPa | Cohesive Force/MPa | Internal Friction Angle/(°) | Tensile Strength/MPa |
|---|---|---|---|---|---|---|---|
| Sandy mudstone | 24 | 2400 | 2.19 | 2.62 | 2.5 | 30 | 0.24 |
| Siltstone 1 | 20 | 2550 | 5.42 | 4.17 | 4.1 | 30 | 0.40 |
| Fine-grained sandstone 1 | 10 | 2500 | 3.64 | 3.04 | 2.8 | 40 | 0.30 |
| Mudstone 2 | 4 | 2300 | 1.33 | 1.13 | 1.0 | 30 | 0.20 |
| Decapodite | 7 | 2500 | 3.27 | 4.68 | 3.6 | 29 | 0.36 |
| 3Coal | 5 | 1500 | 1.18 | 0.53 | 1.0 | 24 | 0.25 |
| Mudstone 1 | 4 | 2300 | 1.33 | 1.13 | 1.0 | 30 | 0.20 |
| Fine-grained sandstone 2 | 16 | 2500 | 4.10 | 3.28 | 3.0 | 28 | 0.33 |
| Fine sandstone 2 | 8 | 2500 | 1.42 | 3.64 | 3.7 | 30 | 0.28 |
| Mudstone 3 | 4 | 2300 | 1.33 | 1.13 | 1.0 | 30 | 0.20 |
| Siltstone 2 | 8 | 2550 | 5.42 | 4.17 | 4.1 | 30 | 0.40 |
| Aquifer | 20 | 1800 | 1.31 | 1.64 | 1.7 | 30 | 0.20 |

*3.2. Analysis of Simulation Results*

3.2.1. Vertical Stress Analysis

Destruction of coal seam floors is most often caused by mining disturbance, which mainly acts on the floor rock stratum in the form of stress transmission, and it is very easy for the floor to be destroyed when stress concentration occurs. According to the numerical simulation, the stress action law of the floor rock stratum was obtained. On this basis, combined with the plastic failure analysis of the back floor rock stratum, the floor damage failure and stress action mechanism can be effectively addressed. According to the numerical simulation results, several groups of vertical stress data at different monitoring positions of the floor were obtained. Because the number of data obtained by simulation is too large, the monitoring data at 0 MPa, 2 MPa, 4 MPa, 6 MPa, 8 MPa, and 10 MPa were selected for analysis. The vertical stress change curves at different mining distances were determined by selecting the stress values at monitoring points A (85, 86, 40) and B (95, 96, 20) (see Figure 8).

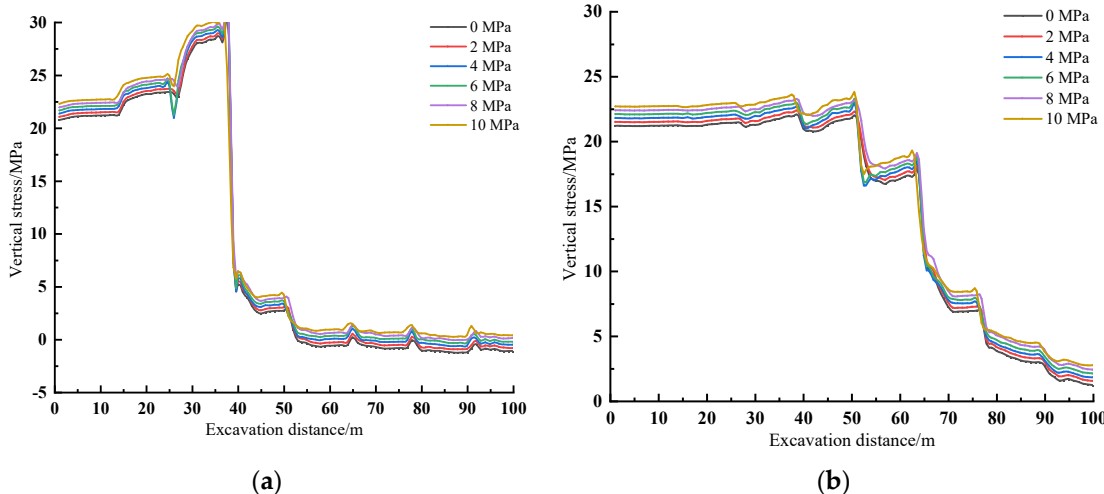

**Figure 8.** Vertical stress curves of monitoring points under different water pressures. (**a**) Monitoring point A, (**b**) monitoring point B.

The vertical stress change curve of monitoring point A reveals an initial increase followed by a decrease with the progression of mining. Specifically, at distances ranging from 0 to 30 m, as the excavation position approaches the location directly above the

monitoring point, the vertical stress on monitoring point A gradually increases until it reaches a maximum value of 30 MPa. Conversely, when the excavation position is far from the monitoring point (40 m or more), the vertical stress on the monitoring point decreases abruptly. The stress concentration occurs at the location of the monitoring point when the excavation reaches it. The magnitude of the vertical stress is positively correlated with the water pressure; however, the influence of water pressure on the vertical stress is limited.

It can be seen from monitoring point B that the monitoring point is subject to a slow increase of vertical stress as the mining proceeds, and the maximum vertical stress at monitoring point B is 24 MPa when the excavation reaches 50 m. As the excavation position moves gradually away from monitoring point B, the vertical stress decreases rapidly. Since the location of monitoring point B is larger than the vertical distance between monitoring point A and the excavation, the vertical stress at monitoring point B is smaller than that at monitoring point A, and the impact of vertical stress at monitoring point B is smaller than that at monitoring point A. Under the action of water pressure, the greater the water pressure, the greater the vertical stress of monitoring point B. After the water pressure reaches 6 MPa, the influence on the vertical stress of the floor rock stratum gradually increases.

### 3.2.2. Plastic Zone Analysis

According to the numerical simulation under the water pressure of 0–10 MPa, the following failure nephogram of the plastic zone was obtained. Since the number of cloud pictures obtained by simulation is large, to facilitate analysis and fully consider the impact of water pressure on plastic failure, the cloud pictures of plastic failure of coal seam floors when excavating to 80 m under water pressure of 0, 2 MPa, 4 MPa, 6 MPa, 8 MPa, and 10 MPa were selected for analysis (see Figure 9).

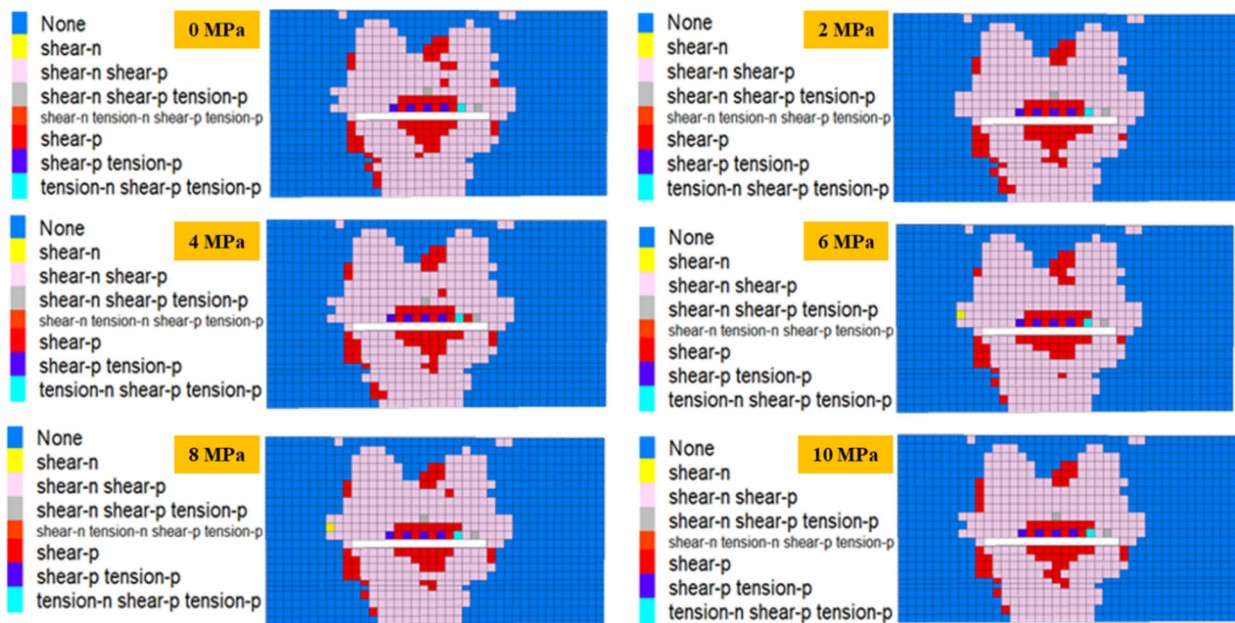

**Figure 9.** Plastic zone under different water pressures.

Water pressure has a certain influence on the disturbance process of coal seam floor mining. With the increase in water pressure, the mechanical failure form of the floor plastic failure changes. The greater the water pressure is, the more shear stress and tensile stress are received near the mining location. With the increase of water pressure, after the water pressure reaches 8 MPa, the range of the plastic failure area is basically unchanged, while the type of plastic failure changes. With the development of mining, the longitudinal and transverse ranges of floor failure are increasing. When the working face is mined to 80 m, the floor damage presents a hemispherical shape. The closer the working face is to the floor,

the larger the damage range is, and the farther away the working face is from the floor, the smaller the damage range is. When the karst water pressure is small, the influence on the floor damage is small. With the increase in water pressure, the damage depth and scope are increasing. When the water pressure is greater than 3 MPa, the bottom plate begins to be affected by strong water pressure. When the water pressure is about 5 MPa, the floor is seriously affected by the water pressure while also being disturbed by mining. After the water pressure reaches 6 MPa, the influence on the plastic failure of floor rock stratum gradually increases.

### 3.2.3. Water Pressure Analysis

According to the simulation of coal seam excavation, the water pressure variation values of all nodes in the model can be obtained. The data of monitoring points A (85, 86, 40) and B (95, 96, 20) at 2~10 MPa were obtained according to the numerical simulation results, and different water pressure change curves were drawn according to the water pressure change values (see Figure 10).

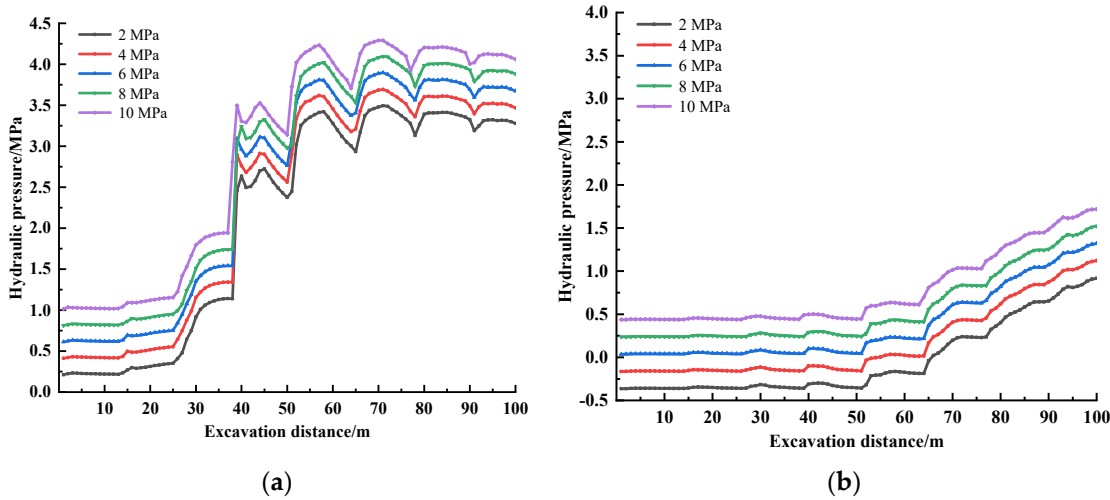

**Figure 10.** Water pressure variation curves at different monitoring points. (**a**) Monitoring point A, (**b**) monitoring point B.

The data presented in Figure 10 demonstrates a positive correlation between the proximity of the monitoring point to the vertical position of the mining location and the level of pore water pressure to which the monitoring point is exposed. This relationship can be attributed to the greater strength of mining disturbance in close proximity to the mining location, which leads to an increase in pore water pressure. The observed increase in pore water pressure can be directly attributed to the heightened degree of mining disturbance in the immediate vicinity of the mining location. The more severe the disturbance is, the faster the water pressure increases. Corresponding to the vertical stress, when the mining location is directly above the monitoring point, the water pressure suddenly increases and then slowly increases until it is stable. The greater the confined water pressure is, the greater the water pressure will be under the same mining disturbance. With exploitation, the water pressure changes little at the initial stage of exploitation, indicating that the exploitation disturbance is weak at this time. When the advancing distance of the working face is about 30 m, the water pressure at monitoring point A starts to change suddenly, and when the advancing distance of the working face is about 50 m, the water pressure at monitoring point B starts to change suddenly. After the water pressure is 6 MPa, the influence on the water pressure change curve gradually increases under the action of mining.

## 4. Prevention Measures

A large number of coal resources are buried in the upper part of the karst aquifer, so it is necessary to take measures to prevent water inrush from the floor to facilitate the mining of this part of coal resources. At present, the prevention measures against karst water inrush from the floor are mainly aimed at the karst water aquifer and the floor aquifers. The method of drainage and depressurization was adopted for the large water pressure in the aquifer, and the method of grouting reinforcement was adopted for the weak aquiclude. Through comparison and analysis of the two control methods, the best control scheme was determined.

### 4.1. Drainage Pressure Reduction

Water drainage and pressure reduction technology refers to the method of drilling holes to drain or reduce the water pressure of the floor aquifer. This method is generally divided into (1) surface drainage and depressurization and (2) downhole drainage and depressurization. This paper studies the problem of water inrush from the floor of deep mining, mainly analyzing the way of dewatering and depressurization. Water pressure is an important factor to be considered for drainage and depressurization. The value of safe water pressure affects the layout position, drainage intensity, and drainage and depressurization time of drainage holes. The safe water pressure is inversely calculated by the formula for the water inrush coefficient, as shown in the following formula:

$$T = \frac{P}{M} \qquad (3)$$

where $T$ is the water inrush coefficient, MPa/m; $P$ is the water pressure, MPa; and $M$ is the water barrier thickness, m.

According to the numerical simulation results, when the water pressure is less than 4 MPa, the impact on mining can be ignored. The relationship between the water inrush coefficient and the safe water pressure obtained is shown in Figure 11, where $T = 0.15$ MPa/m and $M = 30$ m. The safe water head is $P = 4$ MPa, that is, when the water pressure of the aquifer is reduced to 4 MPa by means of hydrophobic pressure reduction (converted to orifice observation, water pressure is about 3.3 MPa). When the following conditions are met, the working face has the basic conditions for safe mining. According to the research of Liu Weitao et al., when the karst water pressure is too large, it is difficult to drill down holes to release water, and it is impossible to carry out hydrophobic pressure reduction [15].

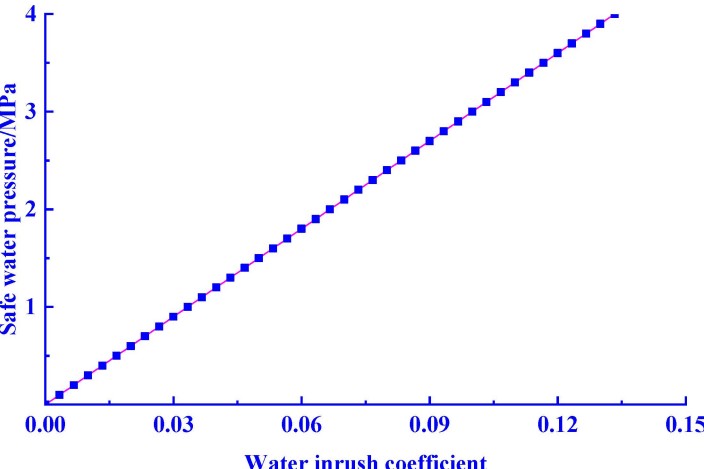

**Figure 11.** Relation between water inrush coefficient and safe water pressure.

### 4.2. Grouting Reinforcement of Bottom Plate and Water-Resisting Layer

There is a thin limestone aquifer under the coal seam floor aquifers, and there is a karst aquifer under the thin limestone aquifer. According to the calculation of rock fluid

solid coupling damage, the water-resisting layer will be damaged and weakened gradually with the introduction of mining disturbance. The thin limestone aquifer will largely interconnect with the karst aquifer, thus forming a water inrush source that threatens the floor. Therefore, the water-resisting layer and the thin limestone aquifer can be reinforced by grouting. According to Liu Weitao's analysis on the data of floor water inrush and engineering practice, under the condition of no structure and gently inclined coal seams, the floor water inrush position of the working face is roughly one-third of the initial pressure line after mining. Mining-induced disturbance of the entire floor rock stratum can lead to the formation of three zones: the floor water conduction failure zone, the complete rock stratum zone, and the confined water conduction height zone. In such cases, the effective thickness of the floor aquifers is likely to experience a significant reduction, which in turn can weaken the water-resistance performance of the aquifer system. In the mining process of the working face, the area where the floor is damaged can be grouted at the same time to improve the water-blocking performance of the water-resisting layer. Generally, when grouting is used for reinforcement, economic costs, slurry ratio effect, and reinforcement effect should be considered.

*4.3. Comprehensive Analysis*

Considering the above two ways to prevent and control the karst water hazard in the floor, detailed exploration of the geological water temperature condition of the mine is required. Determine and select the processing method according to the detection results. The purpose of drainage and depressurization is to pump water, and the purpose of grouting reinforcement is to fill grout. Drainage and depressurization need open drain holes and the use pumps. There is a defect whereby the water pressure is too large for pumping, and the pumps cannot meet the power requirements. Moreover, pumping groundwater also carries the risk of damaging groundwater balance. In the future, the method of grouting reinforcement should optimize the proportion of raw materials and reduce costs in order to promote its use in the field.

**5. Conclusions**

(1) Based on the theory of rock mechanics, the water rock interaction between the lower part of the coal seam floor and the aquifer was analyzed. Based on the triaxial compression tests of saturated and dry rocks, a significant decrease in rock strength in the water-rock coupling effect was obtained; that is, the strength of damage of saturated rocks compared to dry rocks is 36%. The theoretical analysis is consistent with the experimental results and indicates that water-rock coupling seriously damages the stability of the coal seam floor.

(2) According to the simulation results, it is believed that when the water pressure is less than 3 MPa, the vertical stress on the bottom plate does not change much and begins to have an impact on the bottom plate damage. The maximum vertical stress at monitoring point A is 30 MPa, and the maximum vertical stress at monitoring point B is 24 MPa. When the water pressure reaches 6 MPa under the action of mining disturbance, it has a significant impact on the plastic damage of the floor, leading to an increase in the depth and scope of the floor damage, with a maximum damage depth of about 56 m. When the water pressure is greater than 6 MPa, the plastic damage, vertical stress, and node pore water pressure caused in relation to the floor rock gradually increase.

(3) Based on the analysis of two kinds of karst water disaster treatment methods, it is difficult for the drainage and depressurization method to play a role when the water pressure is large. Grouting reinforcement can effectively prevent the occurrence of floor water inrush accidents.

**Author Contributions:** Methodology, L.P.; Software, L.P.; Formal analysis, Q.Z.; Investigation, Q.Z.; Data curation, Q.Z.; Writing—original draft, L.P.; Writing—review & editing, L.P. All authors have read and agreed to the published version of the manuscript.

**Funding:** This research received no external funding.

**Institutional Review Board Statement:** Not applicable.

**Informed Consent Statement:** Not applicable.

**Data Availability Statement:** Not applicable.

**Acknowledgments:** This paper was mainly completed by Qiushuang Zheng and Lifu Pang. Qiushuang Zheng was responsible for the introduction and proofreading of the paper, and Lifu Pang was responsible for the experiment and simulation of the paper.

**Conflicts of Interest:** The authors declare no conflict of interest.

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
