# Peer review of "Research on the Damage Law and Prevention Measures of the Substrate under the Action of Water and Rock"

_water, doi:10.3390/w15081527_

Round 1

Reviewer 1 Report

I suggest this paper for publication after major revisions based on the following comments:

1 In introduction section, please discuss and evaluate the progress of research result rather than listing their research content

2 What are the innovative points of this paper? What are the differences from previous studies? Please explain in the introduction.

3 Figure 5 is replaced by the stress-strain curve, which can reflect more information.

4 What logical connections exist between the second and third chapters, which seem rather isolated.

5 Numerical simulation models usually require validation. How to prove that the numerical simulation model in this paper is correct?

6 What were the specific settings of the rock mechanics parameters in the numerical simulation model?

7 Some relevant literature may be helpful to improve the quality of the article.

Brittleness Evaluation of Coal Based on Statistical Damage and Energy Evolution Theory[J]. Journal of Petroleum Science and Engineering, 2019, 172: 753-763.

Evaluation Method of Rock Brittleness Based on Statistical Constitutive Relations for Rock Damage[J]. Journal of Petroleum Science and Engineering, 2017, 153: 123-132.

Author Response

Reviewer 1

1 In introduction section, please discuss and evaluate the progress of research result rather than listing their research content

[revise1]: Based on the reviewers' comments, the content of the introduction was evaluated and discussed.

2 What are the innovative points of this paper? What are the differences from previous studies? Please explain in the introduction.

[revise2]: According to the reviewers' comments, an explanation of the innovation of the paper is added at the end of the introduction.

3 Figure 5 is replaced by the stress-strain curve, which can reflect more information.

[revise3]: According to the reviewer's comments, redraw Figure 5 as a stress strain curve.

4 What logical connections exist between the second and third chapters, which seem rather isolated.

[revise4]: Explain this based on the reviewer's comments. The second chapter of the paper is devoted to the analysis of water rock interaction from experimental and mesoscopic perspectives. In the third chapter, numerical simulation is used to study the failure of the floor under macroscopic water rock interaction. Chapter 2 and Chapter 3 can support each other.

5 Numerical simulation models usually require validation. How to prove that the numerical simulation model in this paper is correct?

[revise5]: According to the comments of the reviewers, the models in the paper have been established based on relevant geological data, and these models have been widely used in research on water inrush from the roof and floor of coal seams.

6 What were the specific settings of the rock mechanics parameters in the numerical simulation model?

[revise6]: According to the reviewer's comments, the rock mechanics parameters of the numerical model have been added to the paper.

7 Some relevant literature may be helpful to improve the quality of the article.

[revise7]: According to the comments of reviewers, relevant documents have been added to improve the quality of the paper.

Reviewer 2 Report

The review of the manuscript entitled with " Research on the Damage Law and Prevention Measures of the Substrate Under the Action of Water and Rock " has been completed.

This paper studies the damage law of karst water and rock through theoretical analysis and mechanical tests. The mining situation of the working face under different water pressures was analyzed by numerical simulation. It has certain guiding significance for the prevention and control of water inrush in the bottom slab. Overall, I choose "accept with minor revision", the reason stated below.

1. In the first paragraph of the section 2.2 of the manuscript, it is proposed that “The rock specimen is sandstone and processed into a standard cylindrical specimen with a size of 50 * 100 (diameter and height).” The expression lacks units and is not rigorous enough.

2. Figure 4 is not aesthetically pleasing, and it is suggested that the author standardize the size of the text in the figure.

3. In the section 2.2 of the manuscript, the author's analysis of the results of the conventional triaxial tests on dry and saturated rocks is not sufficiently in-depth. As a scientific paper, the test results should not be described only. It is suggested that the author conducts a more in-depth analysis and explain the role of water in this.

4. In the section 3.2.1 of the manuscript, it is proposed that “As the excavation position is far away from the monitoring point, the vertical stress on the monitoring point drops suddenly at 40 m, and the vertical stress on the monitoring point gradually decreases when the excavation position is far away from the monitoring point.” Can you explain the reason for the sudden change?

5. The summary of the conclusions is inadequate and it is suggested that some data results be added.

6. Some of the grammatical structures in the paper are inaccurate and it is suggested that the author check and polish them.

Author Response

This paper studies the damage law of karst water and rock through theoretical analysis and mechanical tests. The mining situation of the working face under different water pressures was analyzed by numerical simulation. It has certain guiding significance for the prevention and control of water inrush in the bottom slab. Overall, I choose "accept with minor revision", the reason stated below.

  1. In the first paragraph of the section 2.2 of the manuscript, it is proposed that “The rock specimen is sandstone and processed into a standard cylindrical specimen with a size of 50 * 100 (diameter and height).” The expression lacks units and is not rigorous enough.

[revise1]: According to the comments of the reviewers, the unit of rock standard rock sample has been added.

  1. Figure 4 is not aesthetically pleasing, and it is suggested that the author standardize the size of the text in the figure.

[revise2]: According to the reviewer's comments, Figure 4 was redrawn.

  1. In the section 2.2 of the manuscript, the author's analysis of the results of the conventional triaxial tests on dry and saturated rocks is not sufficiently in-depth. As a scientific paper, the test results should not be described only. It is suggested that the author conducts a more in-depth analysis and explain the role of water in this.

[revise3]: Based on the comments of the reviewers, an in-depth analysis of section 2.2 of the paper was conducted, and the related role of water was explained.

  1. In the section 3.2.1 of the manuscript, it is proposed that “As the excavation position is far away from the monitoring point, the vertical stress on the monitoring point drops suddenly at 40 m, and the vertical stress on the monitoring point gradually decreases when the excavation position is far away from the monitoring point.” Can you explain the reason for the sudden change?

[revise4]: Explain according to the reviewer's comments. The cause of the sudden change here is due to a decrease in stress. According to the internal and external stress field theory proposed by Academician Song Zhenqi, in the early stage of mining, the internal and external stress field will be formed in the attachment of the open cut. That is, the stress gradually increases on the left side of the open cut, and after reaching the open cut position, the stress begins to decrease in the goaf.

  1. The summary of the conclusions is inadequate and it is suggested that some data results be added.

[revise5]: Based on the reviewers' comments, the conclusions of the paper were re summarized and relevant data results were added.

  1. Some of the grammatical structures in the paper are inaccurate and it is suggested that the author check and polish them.

[revise6]: According to the comments of the reviewers, the grammatical structure and language of the full text have been checked and polished.

Round 2

Reviewer 1 Report

Thanks to the authors for their careful revisions.